# Mitigation of Deoxynivalenol (DON)- and Aflatoxin B1 (AFB1)-Induced Immune Dysfunction and Apoptosis in Mouse Spleen by Curcumin

**DOI:** 10.3390/toxins16080356

**Published:** 2024-08-13

**Authors:** Azhar Muhmood, Jianxin Liu, Dandan Liu, Shuiping Liu, Mahmoud M. Azzam, Muhammad Bilawal Junaid, Lili Hou, Guannan Le, Kehe Huang

**Affiliations:** 1College of Veterinary Medicine, Nanjing Agricultural University, Nanjing 210095, China; azharmuhmood@njau.edu.cn (A.M.); 20196307074@stu.njau.edu.cn (J.L.); t2021062@njau.edu.cn (D.L.); liushuiping@hunau.edu.cn (S.L.); 2018207028@njau.edu.cn (L.H.); 2018107088@njau.edu.cn (G.L.); 2Institute of Animal Nutritional Health, Nanjing Agricultural University, Nanjing 210095, China; 3MOE Joint International Research Laboratory of Animal Health and Food Safety, College of Veterinary Medicine, Nanjing Agricultural University, Nanjing 210095, China; 4Animal Production Department, College of Food and Agriculture Sciences, King Saud University, Riyadh 11451, Saudi Arabia; mazzam@ksu.edu.sa; 5Department of Plant Production, College of Food and Agriculture Sciences, King Saud University, Riyadh 11451, Saudi Arabia; bilawallljunaid@gmail.com

**Keywords:** curcumin, immunomodulation, mycotoxins, apoptosis, cytokines

## Abstract

In the context of the potential immunomodulatory properties of curcumin in counteracting the detrimental effects of concurrent exposure to Deoxynivalenol (DON) and Aflatoxin B1 (AFB1), a comprehensive 28-days trial was conducted utilizing 60 randomly allocated mice divided into four groups. Administration of curcumin at a dosage of 5 mg/kg body weight in conjunction with DON at 0.1 mg/kg and AFB_1_ at 0.01 mg/kg body weight was undertaken to assess its efficacy. Results indicated that curcumin intervention demonstrated mitigation of splenic structural damage, augmentation of serum immunoglobulin A (IgA) and immunoglobulin G (IgG) levels, elevation in T lymphocyte subset levels, and enhancement in the mRNA expression levels of pro-inflammatory cytokines TNF-α, IFN-γ, IL-2, and IL-6. Furthermore, curcumin exhibited a suppressive effect on apoptosis in mice, as evidenced by decreased activity of caspase-3 and caspase-9, reduced expression levels of pro-apoptotic markers Bax and Cytochrome-c (Cyt-c) at both the protein and mRNA levels, and the maintenance of a balanced expression ratio of mitochondrial apoptotic regulators Bax and Bcl-2. Collectively, these findings offer novel insights into the therapeutic promise of curcumin in mitigating immunosuppression and apoptotic events triggered by mycotoxin co-exposure.

## 1. Introduction

Mycotoxin contamination in feed and food sources presents a significant health hazard to both animals and humans. The Food and Agriculture Organization (FAO) reports that approximately 25 percent of the global cereal supply is contaminated with mycotoxins, with an estimated annual incidence of mycotoxin contamination affecting around 1 billion metric tons of food products [1]. The considerable risk of exposure to farmed animals and humans is through the utilization of mycotoxin-contaminated feed or food, which causes complications in the feed and food supply chain [2]. The mixtures of AFB1, OTA, DON, and ZEN were the most common and showed up in almost one-third of raw material feed samples [3]. DON and AFB1 are produced from fusarium and aspergillus, and are frequently found in swine feed [4]. Mycotoxins enter the food chain through various ways, posing a risk for the development of associated diseases upon ingestion [5]. Immunosuppression induced by mycotoxins manifests through mechanisms such as diminished T and B lymphocyte activity, suppression of immunoglobulin and antibody production, reduced complement activity, and damaged function of macrophage effector cells. These immunological alterations contribute to compromised immune responses and susceptibility to infections upon exposure to mycotoxins [6]. The fungi and mycotoxins in fermented feeds and foods are a major issue worldwide, which needs potential applications in the feed industry for the mitigation of mycotoxin pollution [7]. Efforts to mitigate mycotoxin contamination and its adverse health effects remain imperative for safeguarding public health and ensuring food safety [8].

DON, classified as a type B trichothecene mycotoxin, is renowned for its multifaceted impacts on physiological processes in both human and animal organisms. Its effects encompass disruptions in gastrointestinal permeability, inhibition of growth hormone signaling, dysregulation of genes associated with inflammatory pathways, elicitation of immunotoxin responses, contribution to hematological disorders, and alterations in neuroendocrine signaling [9]. The underlying molecular mechanisms driving these pathological alterations involve a spectrum of intricate processes. DON induces oxidative stress, triggering cellular damage. It activates pathways leading to apoptosis and autophagy, crucial mechanisms governing cellular homeostasis and survival. Furthermore, DON influences membrane integrity, perturbing cellular functions. Additionally, DON exerts inhibitory effects on essential cellular processes including RNA and DNA synthesis, as well as transcription and translation mechanisms, disrupting fundamental molecular pathways vital for cellular function and organismal homeostasis [10]. These diverse molecular actions collectively contribute to the wide-ranging physiological disturbances associated with DON exposure in both human and animal populations. Understanding these intricate molecular mechanisms is critical for devising effective strategies to mitigate the adverse health effects of DON contamination.

AFB1 has the potential to induce both acute and chronic toxicity like liver failure, cancer, and disruption of the immune system in different species of animals, including humans [11]. Chronic exposure to aflatoxin has been documented to have a direct immunosuppressive effect on the immune systems, particularly affecting cell-mediated immune responses [12]. The body’s largest peripheral immunological organ is the spleen. Co-exposure to DON and AFB1 shows synergetic effects, induces hepatic dysfunction, and ultrastructural changes may also occur [13]. The susceptibility to infectious diseases is increased by these factors taken together [14]. The combination of DON and AFB1 could increase the production of reactive oxygen species (ROS) and promote apoptosis in rat hepatocytes, mediated by the apoptosis pathway shared between DON and AFB1 [15]. As compared to single-mycotoxin toxicity, a DON and AFB1 mixture indicates strong toxicity in embryonic zebrafish [16]. However, the effects of a combination at low level of DON and AFB1 on the immunity of mice remain unclear. It is important to understand the synthesis, toxicity, and molecular mechanism of AFB1 and DON to prevent contamination and conduct appropriate detoxification. Several natural remedies or active compounds, such as grape seed, lycopene, sodium selenite, and sporoderm-broken Ganoderma lucidum spores, have been found to be beneficial to mitigate immune inhibition and apoptosis caused by AFB1 and DON [17,18].

Curcumin, the principal bioactive compound derived from the rhizome of *Curcuma longa*, is cultivated extensively across regions including China, India, and various parts of Asia, serving diverse purposes. Comprising the bulk of commercially available curcumin powder are three primary constituents: difurylmethane, commonly recognized as curcumin, constituting approximately 82%; its derivative desmethoxycurcumin (DMC), accounting for around 15%; and bisdemethoxycurcumin (BMDC), comprising approximately 3% of the total composition [19]. Research investigations have unveiled a myriad of notable characteristics attributed to curcumin. Its pharmacological properties encompass potent antibacterial, anticancer, antiviral, and antioxidant attributes [20]. These properties render curcumin a promising candidate for therapeutic applications across various biomedical domains, offering potential avenues for combating microbial infections, malignancies, viral diseases, and oxidative stress-induced pathologies. The multifaceted bioactivity of curcumin underscores its significance in pharmacological research and therapeutic development, warranting further exploration into its mechanisms of action and clinical applications [21].

Curcumin also has tissue- and organ-protective and immunomodulatory effects [22,23]. It has been described in previous research that curcumin prevents BDE-209-induced immunosuppression in broilers [24]. It has been demonstrated that curcumin is effective at reversing changes in immunoglobulin and gene expression levels associated with AFB1-induced immunosuppression in the spleen of ducklings [25]. Curcumin inhibits cell apoptosis by restoration of pro-apoptotic and anti-apoptotic proteins, reducing the Bax/Bcl-2 ratio, limiting mitochondrial outer membrane permeabilization, decreasing cytochrome-C release, and improving cell apoptosis in response to external or internal stimuli [26,27,28]. Moreover, B and T cell stimulation and proliferation suggested that curcumin has a positive effect on the immunological functions of broiler chicken [29]. Curcumin mediates several processes like restoration of CD4^+^ and CD8^+^ T cell populations, reversal of type-2 cytokine bias, reduction in Treg cell populations, and suppression of T cell apoptosis [30]. Previous studies have focused on evaluating the effect of curcumin on individual toxins, with limited exploration of the effects on co-exposure of DON and AFB1. To address the existing research gap, our current study aimed to investigate the protective role and immunostimulatory effects of curcumin in mitigating the combined effects of DON and AFB1 on the immune system. This was achieved by assessing immunological functions and examining mRNA and protein expression levels of mitochondrial apoptosis-related factors to understand the influence of curcumin on the alleviation of combined AFB1- and DON-induced immunotoxicity in mice. The present investigations may provide useful information on the possible application of curcumin as a therapeutic approach for reducing the combined toxic effects of AFB1 and DON on the immune system in humans and animals.

## 2. Results

### 2.1. Curcumin Restored Microscopic Structural Changes in Spleen

Microscopic changes in the spleen of mice treated with AFB1, DON, and curcumin in different combinations are shown in Figure 1. The Figure 1A control group exhibited a compact and well-organized structure, characterized by a distinct nucleus. Additionally, the demarcation between the red and white pulp of the spleen was evident and clearly defined. Following the application of DON + AFB1 as shown in Figure 1B, there were noticeable changes in the arrangement of spleen cells and the structure of red and white pulp. Specifically, the boundaries between the red and white pulp became unclear, being severely damaged. Moreover, the trabeculae in the spleen became more tightly packed, while the number of lymphocytes significantly decreased compared with the curcumin-treated group. Contrarily, in the curcumin group as shown in Figure 1C the red pulp and white of the spleen became more distinct and clearer as compared to the DON + AFB1 group. Similarly as in the Cur + DON + AFB1 group, the original structure of the spleen tissue is gradually restored, and the above-mentioned pathological changes are alleviated, as shown in Figure 1D.

### 2.2. Effects of Curcumin on Mice Exposed to DON + AFB1 Combination

The action of curcumin and DON + AFB1 on immunoglobulin production in the serum of control and experimental mouse groups is depicted in Figure 2. Our findings indicate a significant decrease in IgA and IgG levels in the DON + AFB1-co-exposed treated group as compared to the control group, as shown in Figure 2A,B. On the contrary, the group treated with curcumin alone and in combination with DON + AFB1 exhibited a significant upregulation in the production of IgA and IgG respectively, in the serum of mice when compared to the toxin concentration-challenged group. Additionally, the mRNA levels of cytokines, including *TNF-α*, *IFN-γ*, *IL-2*, and *IL-6*, were decreased in the DON + AFB1-treated group as compared to untreated control and curcumin group, as shown in Figure 2C–F. To provide an overview of the impact of curcumin therapy on the immune system of mice compromised by DON + AFB1, herein the number of CD4^+^ and CD8^+^ T cells is investigated, as well as the ability to impair cytokine levels in the mouse spleen. As shown in Figure 2G–J, DON + AFB1 exposure significantly reduced the percentage of CD4^+^ and CD8^+^ T cells in the spleen of mice as compared to the control and curcumin-treated groups (Figure 2G,H). Meanwhile, the ratio of CD4^+^/CD8^+^ also decreased in the combined toxin-exposed group as compared to control and curcumin-treated groups. However, the curcumin treated against DON + AFB1 group had significantly increased percentages of CD4^+^ and CD8^+^ T cells and restored cytokine levels (*TNFα*, *IFNγ*, *IL-2*, and *IL-6*) that were downregulated by DON + AFB1 exposure.

### 2.3. Curcumin Protects against DON + AFB1-Induced Spleen Cell Apoptosis of Mice

The effects of curcumin and co-exposure to DON and AFB1 on splenocyte apoptosis are illustrated in Figure 3. The *Bcl-2* mRNA expression levels were significantly reduced in the DON + AFB1 group compared to the control group (Figure 3A). Conversely, there was a notable upregulation in *Bcl-2* mRNA levels in the group treated with curcumin. Figure 3B–D depict the effects of DON + AFB1 and curcumin on the relative mRNA levels of Bax, Caspase-3, and cytochrome-C in the control and experimental groups. Compared to the control group, the DON + AFB1-treated group showed significantly increased mRNA levels of *Bax*, *Caspase-3*, and *Cyt-c*. However, curcumin supplementation in the toxin co-exposure group reversed the upregulation of *Bax*, *Caspase-3*, and *Cyt-c* mRNA levels compared to the DON + AFB1-treated group.

To investigate the protective effects of curcumin on DON + AFB1-induced apoptosis, we assessed the protein expression levels of *Bcl-2*, *Bax*, *Caspase-3*, and *Cyt-c* in the spleen of mice using Western blotting (Figure 3E–J). The protein expression trends mirrored those observed at the mRNA level. Co-treatment with DON + AFB1 resulted in a significant upregulation of *Bax*, *Caspase-3*, and Cyt-c protein levels, while the protein expression level of Bcl-2 was notably downregulated compared to the control and curcumin-treated groups. Additionally, the co-exposure to DON + AFB1 significantly increased the apoptosis rates of splenocytes (Figure 3K,L) compared to the control and curcumin-treated groups. However, pretreatment with curcumin significantly decreased the apoptosis rates in the combined toxin-treated group.

## 3. Discussion

DON and AFB1 contaminate both animal feed and food products, imposing a threat to both public health and the livestock sector, exclusively in developing countries with poor feed and food handling practices that favor mold growth. Regardless of various studies investigating the neutralization of harmful effects exerted by DON and AFB1, practical and cost-effective solutions are still lacking. Several therapeutics are impractical, costly, or leave toxic residues, compromising human and animal health. However, over the past few decades, phytochemical agents and bioactive compounds of plants have proven effective in neutralizing mycotoxins’ hazards with minimal side effects.

The immunopathological disorders induced by DON constitute a thoroughly documented aspect of its profound toxicity, whereas AFB1 stands out as the predominant aflatoxin contaminant, significantly compromising the safety of animal feed and human health [31,32]. Therefore, exposure to DON and AFB1 has hidden risks to food and animal feed. However, previous investigations on DON and AFB1 have mostly focused on their single toxicity on different body organs, while their combined toxicity remains largely unknown [33,34].

Mycotoxins’ combined exposure has previously been found to impair the spleen and lead to immune inhibition and apoptosis [35]. In the immune system, the spleen is one of the most important secondary lymphoid organs, which contains the white and red pulp affected by DON and AFB1 toxicosis [36]. The white pulp is mainly involved in immune functions, while the red pulp is primarily responsible for filtering and processing blood components and playing a crucial role in capturing and destroying harmful pathogens and maintaining the immune system [36]. The combined effect was explored by subjecting mice to DON and AFB1 mixtures, indicating that the combination of these two aflatoxins produces severe toxicity [37]. Therefore, spleen injury by any toxic element is often linked to compromised immune function and increased susceptibility to infections [38].

Curcumin, a polyphenolic compound obtained from turmeric (*Curcuma longa*), has gained the attention of researchers due to its multifaceted pharmacological properties, including antioxidant, immune-modulatory, anti-inflammatory, and anti-tumor properties [31]. Curcumin has attracted a great deal of attention as a novel and green feed additive in recent years. However, the effects of curcumin on the immune response induced by the combination of DON and AFB1 still need a comprehensive study. Keeping in view the enormous health benefits of curcumin, we explored the protective effects of curcumin against the combined toxic effects of DON and AFB1 on the immune system of mice.

Our findings manifested that a DON and AFB1 combination treatment induced histopathological changes, decreased lymphocyte count, and induced red pulp congestion which inhibited the developmental status of the spleen and caused spleen injury, while the curcumin treatment mitigated these adverse effects; these results are in accordance with previous findings in which similar positive effects have been reported in broilers [39]. Another study on broilers showed that on the 28th day, AFB1 induced severe congestion in the spleen red pulp area and a reduction in lymphocytes but low and medium doses of curcumin showed interventional effects on these pathological changes in the spleen [40], whilst our study showed that curcumin could mitigate the DON and AFB1 combined toxicity in the spleen of mice. Furthermore, spleen injury was ameliorated by curcumin treatment, indicating that it effectually alleviates immunosuppression in the spleen of mice exposed to DON and AFB1 combination.

The exposure to the DON and AFB1 combination primarily impaired cell-mediated immunity [37]. Thus, to further confirm the ameliorative effects of curcumin, the concentration of immunoglobulins, the percentage of CD8^+^ and CD4^+^ T lymphocytes, along with the cytokines related to cellular immunity was determined. The spleen serves as a site for T lymphocyte interaction and activation with other immune cells found on the exterior of cell surface proteins [41]. CD4^+^ lymphocytes are commonly referred to as T helper cells based on the expression of glycoproteins on their cell surface, which interact with major histocompatibility complex class II (MHC-II) molecules. Furthermore, CD4^+^ cells are divided into different Th1 and Th2 subsets based on their cytokine production and functions. The Th1 cells secrete *TNFα*, *IFNγ*, *IL-2*, and *IL-6* cytokines, which promote cell-mediated immune responses [42]. CD8^+^ cells, also known as cytotoxic T cells, can recognize infected or abnormal cells and secrete different cytokines and chemokines to destroy them [43]. In addition to their cytotoxic effects, the production of cytokines helps to activate the immune response and stimulate cells to remove the toxic effects [44]. Therefore, T lymphocyte subsets and cytokine levels provide insights into cellular immunity. Previous studies have shown that the DON and AFB1 co-exposure involved an abnormal ratio and levels of T lymphocytes and cytokine-induced immune dysfunction [15,45]. Thus, our results demonstrated that the DON and AFB1 combination decreased immunoglobulin IgA and IgG levels as well as CD4^+^ and CD8^+^ T lymphocyte subsets, and the mRNA expression level of *TNFα*, *IFNγ*, *IL-2*, and *IL-6*. Our results are in accordance with a previous study which demonstrated that a DON and AFB1 combination is considered to be immunosuppressive [37]. We observed similar results in our previous in vitro and in vivo studies, which set out to check the immunosuppressive effects of DON and AFB1, which reduced the expressions of *TNFα*, *IFNγ*, *IL-6*, and *IL-2* mRNA [37]. Furthermore, previous studies have indicated that the combined toxicity of DON and FB1 in piglets could result in impairment of T lymphocytes, hence decreasing levels of *TNFα*, *IFNγ*, and *IL-2* [46]. In addition to that, our current study showed that supplementary curcumin enhanced the level of immunoglobulins IgA and IgG, CD4^+^ and CD8^+^ T lymphocyte subsets, and cytokines to different degrees, to make it clear that curcumin has beneficial effects on immune functions. These findings were observed in a previously investigated study [40], which indicated that curcumin could activate B lymphocytic proliferation and differentiation to produce [47] plasma cells that synthesize and secrete antibodies against AFB1-induced spleen injury in broilers [48]. Some previous studies have shown that curcumin can restore the population of CD4^+^ and CD8^+^ cells, suggesting a potential for re-establishing immune balance [49]. In general, curcumin can promote the growth and development of plasma cells in immune organs to resist DON- and AFB1-induced immune system injury to some extent. Cytokines, such as *TNFα*, *IL-2*, *IFNγ*, and *IL-6*, are released by T lymphocytes (CD4^+^ or CD8^+^) which can regulate the cell activity and play a crucial role in cellular immunity in broilers [50,51].

Apoptosis, often referred to as a programmed form of cell death, is a highly regulated and controlled process, regulating cell numbers and the cell division rate during both normal functioning and pathological conditions [52], whereas excessive elimination of dead cells is interconnected with immunosuppression [53]. The use of the DON and AFB1 combination showed significant effects on cellular processes, as mitochondria are considered the main target of toxicosis [54]. In the process of apoptotic signal transduction of mitochondrial damage, changes occur in the permeability of mitochondrial cell membranes, which provide a channel for Cyt-c to transfer into the cytoplasm; Cyt-c interacts with apoptosis protease activating factor (Apaf-1) and pro-caspase-9, leading to the auto-processing of caspase-9. The activated caspase-9, in turn, directly activates caspase-3, resulting in its transformation into an active cleaved form known as cleaved caspase-3. This sequence of events ultimately initiates mitochondrial-mediated apoptosis [55,56]. Additionally, the mitochondrial-mediated apoptotic pathway is intricately regulated by key members of the Bcl-2 family, which surrounds proteins with pro-apoptotic and anti-apoptotic functions. Anti-apoptotic and Bcl-2 proteins in mitochondria inhibit the release of Cyt-c into the cytoplasm by impairing the mitochondrial permeability transition pore. Conversely, a pro-apoptotic protein called Bax is found as a cytoplasmic monomer and loosely associates with mitochondrial outer membranes under normal conditions. Upon stimulation, Bax translocates to the mitochondria, where it collaborates with Bcl-2, promoting the release of Cyt-c and facilitating apoptosis.

The delicate balance between Bcl-2 and Bax levels is pivotal, as an imbalance can contribute to the promotion of apoptosis in cells [57]. In our current investigations, using the DON and AFB1 combination led to an elevation of caspase-3, caspase-9, Cyt-c, and Bax expression levels, while decreasing the Bcl2 expression level. Our findings are consistent with results reported by the authors of [37], who observed upregulation of Cyt-c, caspase-3, caspase-9, and as well as downregulation of Bcl2 by using a DON and AFB1 combination. Additionally, curcumin treatment decreased the Cyt-c, caspase-3, caspase-9, and Bax mRNA and protein expression levels while increasing the Bcl2 expression level. Our results aligned with a previous study reported by the authors of [24], who observed that curcumin decreased the expression of caspase-3, Bax, and Cyt-c as well as increased Bcl2 in BDE-209-induced apoptosis in broilers.

However, curcumin treatment effectively reversed the pathological conditions by modulating the mitochondrial-mediated apoptotic pathway in the spleen of mice. Therefore, anti-apoptosis is one of the best curcumin-induced mechanisms against DON and AFB1’s combined toxicity.

## 4. Conclusions

In conclusion, we found that the toxicity of the DON and AFB1 combination could induce immunosuppression and apoptosis in the spleen of mice. Additionally, we found that curcumin could decrease the co-exposure toxicity of DON and AFB1 by activating a series of defense responses. Curcumin could be used as a feed additive to protect the immune system. Further studies are needed to enhance its effectiveness using effective doses and proper delivery to the target organ of the animals.

## 5. Materials and Methods

### 5.1. Animals and Experimental Design

The Committee for the Care and Use of Experimental Animals at the Nanjing Agricultural University (Animal Ethics Number: NJAU.No20210624093) approved all the experimental animals. C57BL/6 male mice were bought from the GemPharmatech Co., Ltd. (Nanjing, China). and kept under standardized environmental conditions during the whole experimental period.

After an adaptive rearing period, sixty male mice (*n* = 15) were randomly assigned to four groups; (1) control group (receiving basal diet only); (2) DON + AFB1 group (received DON + AFB1 at a dose of 0.1 mg/kg + 0.01 mg/kg, respectively) (purity ≥ 99.8%) (Pribolab, Qingdao, China); (3) curcumin group (received curcumin at a dose of 5 mg/kg BW) (purity ≥ 98.0%); and (4) DON + AFB1 + curcumin group (received DON + AFB1 + curcumin at a dose of 0.1 mg/kg + 0.01 mg/kg + 5 mg/kg BW, respectively, via oral gavage once a day in distilled water). The experiment was conducted continuously for 28 days.

### 5.2. Sample Collection

After 28 days, all animals were euthanized by CO_2_ inhalation, and blood samples were collected from the retro-orbital sinus. The process of enucleation of the eye made blood collection easier. The spleens were removed right away, dried with filter paper, rinsed in ice-cold saline, and then weighed. For histopathological analysis, a portion of the spleen was preserved in 10% neutral buffered formalin, and another portion was used to create a homogenate for related kit detection. Furthermore, a portion was prepared in a single-cell suspension using an earlier-developed technique for flow cytometry examinations. To assess the mRNA expression of Caspase-3, Caspase-9, Bcl-2, Bax, and Cyt-c, the rest of the spleen was rapidly frozen in liquid nitrogen and stored at −80 °C for protein and RNA extraction.

### 5.3. Histopathological Examination

A cross-sectional slice of spleen from each of the four groups was subjected to 72 h of fixation in 10% neutral buffered formalin for 72 h following standard histological processing. Following paraffin embedding, the spleen samples were sectioned coronally at a thickness of 5 μm, and then subjected to deparaffinization and rehydration processes. Finally, the slides were stained with hematoxylin and eosin (H&E) to assess tissue structure and then subjected to a digital microscopy scanner (Panoramic MIDI, equipped with a 20× microscope objective from 3D Histech Ltd., Budapest, Hungary) for scanning.

### 5.4. Detection of Immunoglobulins (Igs) through ELISA

The production of Igs, including immunoglobulin A (IgA) (Cat. No. H108) and immunoglobulin G (IgG) (Cat. No. H106) in the mouse serum was measured using ELISA kits (purchased from Nanjing Jiancheng Bioengineering Institute, Nanjing, China) following the manufacturer’s protocols.

### 5.5. Splenic T Lymphocyte Subset Analysis

To assess the proportions of CD4+ and CD8+ T lymphocytes, single-cell suspensions from the spleen were analyzed using FACScan flow cytometry. The spleen was first placed in a culture dish with 6 mL of cold phosphate-buffered saline (PBS) and then dissected into small fragments. Splenic cells were isolated by gently tapping these fragments on a sterile 200-mesh stainless-steel sieve using an inoculator. The collected cells were subjected to centrifugation (1000 rpm, 10 min) and treated with red blood cell lysis buffer (Beyotime, Shanghai, China). Following two washes with cold PBS, the cells were resuspended in cold PBS to a final concentration of 2 × 10^5^ cells/mL. For flow cytometry analysis, 100 μL of cell suspension was stained with fluorescein isothiocyanate (FITC)-conjugated anti-mouse CD4 antibody, phycoerythrin (PE)-labeled anti-mouse CD4 antibody, and conjugated anti-mouse CD8 antibody (MultiSciences, Hangzhou, China). The stained cells were analyzed using an FAC scan flow cytometer (Becton Dickinson, Franklin Lakes, NJ, USA). For fluorescence detection, the excitation and emission wavelengths for FITC were 488 nm and 525 nm, respectively, and for PE, they were 488 nm and 575 nm. FlowJo software v10 (Tree Star Inc., Ashland, OR, USA) was utilized for data analysis.

### 5.6. Analysis of Spleen Apoptosis

To assess apoptosis rates, on day 28 of the experiment, fifteen mice from each group were euthanized and the spleen samples were collected to determine the proportions of apoptotic cells via flow cytometric analysis. Immediately after excision, the spleen was homogenized to produce a cell suspension and then filtered through a sieve. Then, the cells were subsequently washed and resuspended in 1× binding buffer at a concentration of 1 × 10^6^ cells/mL. Annexin V-fluorescein isothiocyanate (V-FITC, 5 µL) and propidium iodide (PI, 5 µL) were added to the 100 µL cell suspension, and then incubated at 25 °C for 15 min in the absence of light. Subsequently, 400 µL of 1× binding buffer (400 µL) was added to the mixture, and apoptotic cells were then analyzed by flow cytometry within 1 h. The annexin V-FITC apoptosis detection kit (Cat. No. C1067 Beyotime, China) was used and finally, the apoptosis rate was assessed by Becton Dickinson (BD, USA) flow cytometer.

### 5.7. Quantitative Real-Time PCR Analysis

Total RNA was extracted from the mouse spleen using the Trizol reagent (Takara, Shiga, Japan), following the recommended protocol. The RNA concentration was measured by spectrophotometer (Gene Quant 1300 GE, Marlborough, MA, USA) at 260 nm and 280 nm. RNA samples with 260/280 nm absorbance ratios between 1.8 and 2.1 were considered to have acceptable quality and concentration for subsequent analyses. Transcript First-Strand cDNA Synthesis Super Mix (Roche, Basel, Switzerland) was used to synthesize complementary DNA (cDNA), adhering to the recommended instructions. Primer sequences are detailed in Table 1. PCR was conducted on a 7000 real-time PCR detection system (ABI, Boston, MA, USA) with SYBR Green PCR Core Reagents (Roche, Switzerland). The housekeeping gene *GAPDH* served as a reference for normalization, and data analysis was performed using the 2−ΔΔCt method. The results were expressed as relative mRNA levels.

### 5.8. Relative Expression of Proteins by Western Blot Analysis

A commercial kit (C3606, Beyotime, China) was used for protein extraction, while whole proteins were extracted using super active (RIPA) Radio immunoprecipitation assay lysis buffer (P0013B, Beyotime, China). The protein concentrations were estimated via a BCA analysis kit (P0012S, Beyotime, China). Samples in the range of 30–50 μg underwent analysis through 12% sodium dodecyl sulfate–polyacrylamide gel electrophoresis and were subsequently transferred onto PVDF membranes (Millipore, Burlington, VT, USA). The membranes then were incubated at 4 °C (overnight) with primary antibodies in primary antibody diluents recognizing Cyt-c (1:800 Wanleibio, Wuhan, China), β-actin (1:1000 ZSGE-BIO, Beijing, China), Bax (1:1500 ABclonal, Shenyang, China), Bcl-2 (1:500 Wanleibio, China), and cleaved caspase-3 (1:500 Wanleibio, China). The primary antibodies were then probed with goat anti-rabbit IgG and goat anti-mouse IgG (ZSGE-BIO, China) at 37 °C for 2 h. The protein bands for Bcl-2, Cyt-c, Bax, and cleaved Caspase-3 were detected using an enhanced ECL reagent (Beyotime, China), with β-actin serving as a control. Quantitative analysis was conducted using an Image Quant LAS 4000 (GE Healthcare Life Sciences, Marlborough, MA, USA).

### 5.9. Statistical Analysis

Statistical analyses were conducted using one-way ANOVA. The least significant difference (LSD) test was applied to identify differences between treatment groups. A *p*-value < 0.05 was considered statistically significant. Results were presented as mean ± SD. Statistical analyses were performed using SPSS software version 22 (IBM Corporation, Armonk, NY, USA).

## Figures and Tables

**Figure 1 toxins-16-00356-f001:**
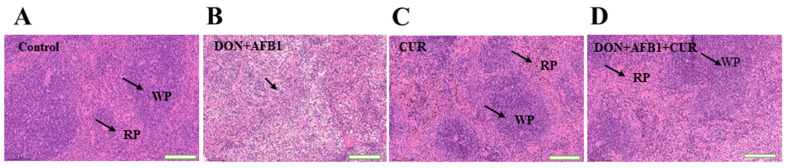
Microscopic changes in mouse spleen treated with DON, AFB1, and curcumin. The image magnification is 50×; WP, white pulp; RP, red pulp; black arrows indicate the splenic demarcation between white pulp and red pulp, (**A**) represents well-organized structure and distinct nucleus, (**B**) represents unclear boundaries between white pulp and red pulp, (**C**) represents distinct and clear margins between white pulp and red pulp, and (**D**) represents increased lymphocytes and evident margins as compared with the DON + AFB1 group.

**Figure 2 toxins-16-00356-f002:**
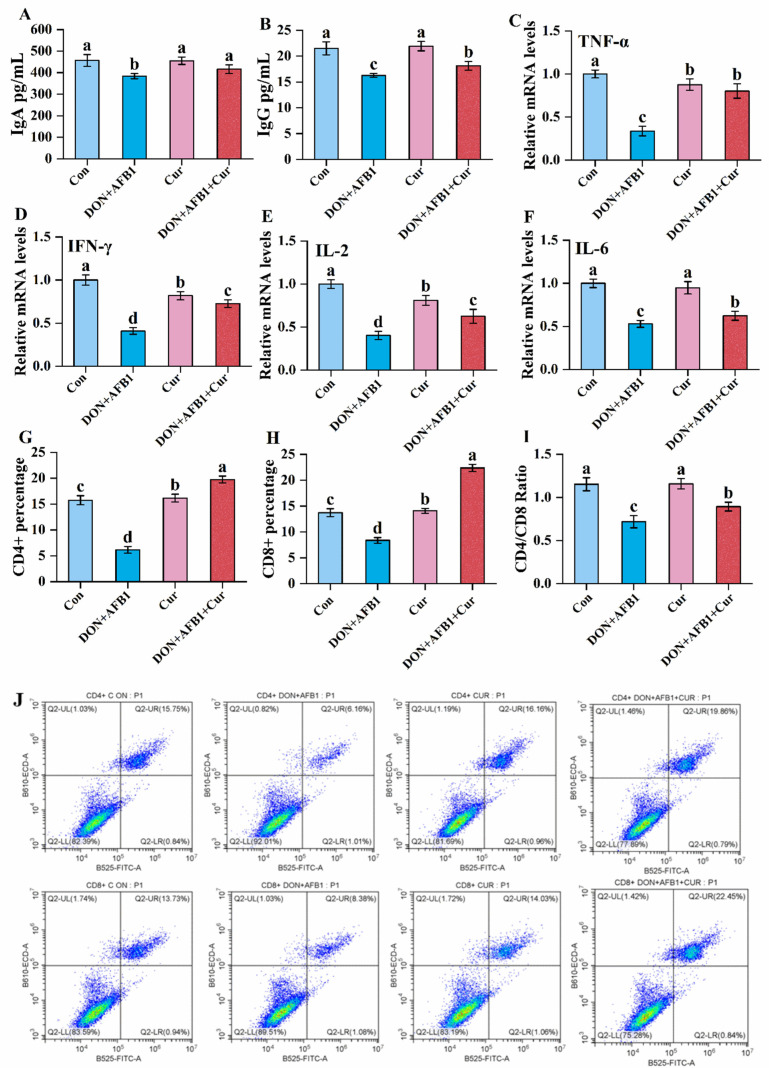
Effects of curcumin on the immune system of mice exposed to DON + AFB1. (**A**,**B**) IgA and IgG were detected by an ELISA KIT in the serum of mice. (**C**–**F**) The mRNA levels of *TNF-α*, *IFN-γ*, *IL-2*, and *IL-6* cytokines in the spleen tissue of mice were quantified by qRT-PCR. (**G**,**H**) The percentage of CD4^+^ and CD8^+^ T lymphocytes were assessed by flow cytometry, and the CD4^+^/CD8^+^ ratio is presented (**I**). (**J**) The representative CD4^+^ and CD8^+^ flow cytometry profiles of splenic cells are shown. The data are represented as the means ± standard deviation (*n* = 15). Different lowercase letters represent *p* < 0.05.

**Figure 3 toxins-16-00356-f003:**
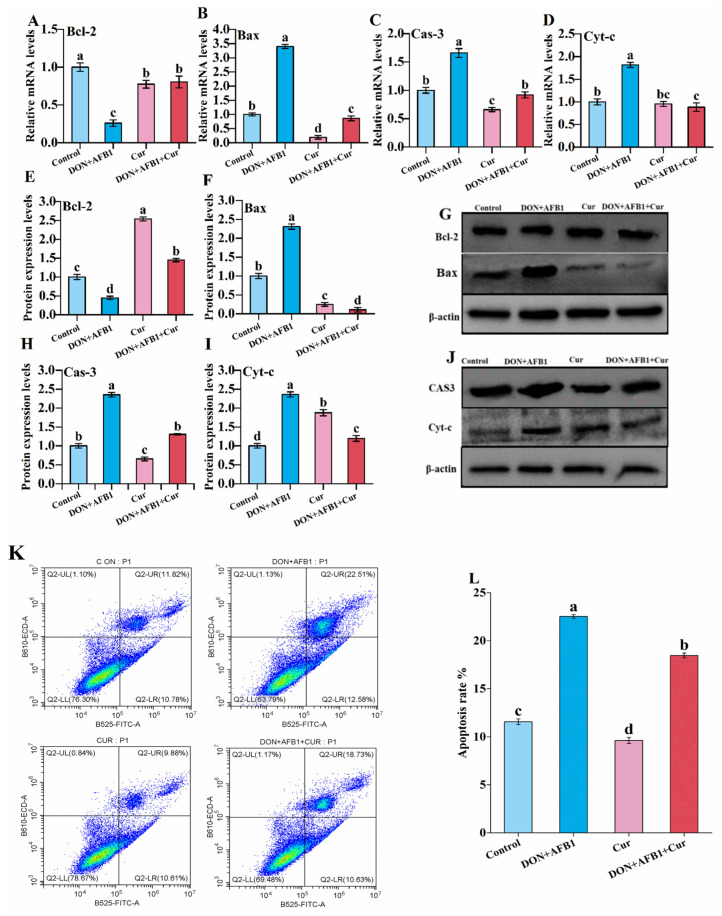
Curcumin treatment restrains the DON + AFB1-induced mitochondrial-mediated pathway in the spleen tissue of mice. (**A**–**D**) The mRNA expression of *Bcl-2*, *Bax*, *Cas-3*, *Cyt-c* was quantified by qRT-PCR. (**E**–**J**) The protein expression of Bcl-2, Bax, Cas-3, Cyt-c was measured by Western blotting. The lower bands represent β-actin for all above-validated proteins. (**K**) The splenocytes’ apoptosis was analyzed by flow cytometry. (**L**) Quantitative analysis of the apoptosis rate. The results are shown as the means ± standard deviation (*n* = 15). Different lowercase letters represent *p* < 0.05.

**Table 1 toxins-16-00356-t001:** Primers used for quantitative real-time PCR analysis.

Gene	Primer Sequences	Gene Numbers
*IL-2*	For 5′ CCAAGCAGGCCACAGAATTG 3′	NM_008366.3
	Rev 5′ GCTGACTCATCATCGAATTGGC 3′	
*IL-6*	For 5′ CCAGGAACCCAGCTATGAAC	NM_204628.1
	Rev 5′ CTGCACAGCCTCGACATT	
*IFN-γ*	For 5′ ACGGCACAGTCATTGAAAGC 3′	NM_008337.4
	Rev 5′ TCACCATCCTTTTGCCAGTTC 3′	
*TNF-α*	For 5′ CGTCGTAGCAAACCACCAAG 3′	NM_013693.3
	Rev 5′ TTGAAGAGAACCTGGGAGTAGACA 3′	
*GAPDH*	GCATCTTCTTGTGCAGTGCC	NM_008084.4
	TACGGCCAAATCCGTTCACA	

*IL-2*: Interleukin-2, *IL-6*: Interleukin-6, *IFN-γ*: interferon-gamma, *TNF-α*: Tumor necrosis factor-α, *GAPDH*: Glyceraldehyde 3 phosphate dehydrogenase.

## Data Availability

Data supporting the conclusions of this study can be made available upon request from the corresponding author.

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
