# Peer review of "Mitigation of Deoxynivalenol (DON)- and Aflatoxin B1 (AFB1)-Induced Immune Dysfunction and Apoptosis in Mouse Spleen by Curcumin"

_toxins, 2024, doi:10.3390/toxins16080356_

Round 1

Reviewer 1 Report

Comments and Suggestions for Authors

Overall, the manuscript is well written and the experimental procedures were well designed. However, the following items should be addressed to improve its quality:

-Always use subscript “1” after “B” when referring to “ B1” or “AFB1”.

-Section 2, subsection 1.1.1.1. (check the numbers of all sections): Figure 1A shows the control group, not the spleen of mice treated with AFB1, DON, and curcumin. Also, letters A, B, etc in the legends of Figure 1 do not match with the contents of this figure.

-Section 4: The discussion can be more informative about the main outcomes from the study.

-Section 5.1. Animals and Experimental Design: The description of the treatment groups is confusing, please indicate clearly which group received each treatment and for how long. In addition, put the rationale for using such doses of DON, AFB1 and curcumin. Importantly: there is no reference on ethics approval of the experiment.

Author Response

"Please see the attachment".

Reviewer 2 Report

Comments and Suggestions for Authors

The study investigated the mitigating effect of curcumin on the immune system and apoptosis in the spleen at combined exposure of DON and AFB1.

The paper is well-written and contains most of the relevant information. It contains novelties because such an investigation has not been published in mice.

The major problem with the paper is that individual and combined effects cannot be evaluated due to the lack of adequate information about the individual effects in mice.

The Introduction contains the most relevant information, but previous studies with combined DON and SFB1 treatment should be mentioned.

par. 1. L 13: It should be mentioned that besides food, feed safety also has importance

par 3 L 9-11: Mycotoxin mitigation effect of organic selenium compounds is much higher than sodium selenite.

p. 8. par 1: This paragraph should move to the right place because it is not the best here.

The Discussion should be rephrased because there is no information about the individual effects of the two mycotoxins and their combination. For this reason, whether the effect induced by the DON or AFB1 and their combined effect are synergistic or antagonistic is not known.

par 1. L 4-7: Some previous studies have examined the combined effects of DON and AFB1; therefore, relevant references should be added.

par. 3. L 1-2: The cited reference is irrelevant because there is no data about the spleen.

par. 3. L 2-4: Relevant references should be added to this statement.

par 3. L 7-8: The cited reference is irrelevant because DON and FB1 were not studied, but AFB1 and OTA co-exposure were.

par 3 L 9-11: The cited study investigated the individual effect of AFB1 but not in combination with DON; therefore, it is irrelevant.

par 3 L 14-15: This sentence has no meaning.

p. 9. L 1-3: The cited study did not show the combined effect of DON and AFB1; therefore, it is irrelevant.

p. 9. L 6-7: The cited study showed the individual but not the combined effect of DOA and AFB1.

p. 9. L 10-13: The cited study showed only the individual effect of AFB1. It is well known, but it does not explain the results of the present study.

p. 9. L 19: The effect of curcumin is correct, but the second part of the sentence did not explain the results of the present study because it is about the individual effect of AFB1.

p. 9. L 21-23: This sentence is correct but irrelevant in this study due to differences in the avian and mammalian immune systems.

The Materials and Methods chapter is correct but requires some corrections.

5.1. Vehicle for the administration of mycotoxin mixture and curcumin should be added.

         It should be mentioned that the mode of application was daily oral gavage.

5.2. L 1-2: Please describe that blood samples were taken before or after euthanasia.

The Conclusions are acceptable.

The Reference list should be formatted according to the Instructions of the Authors. In some cases, the names of the journals are missing.

Reviewer 3 Report

Comments and Suggestions for Authors

Dear authors

Happy day

The paper is fine but need some improvement

In the introduction part

1-      Kindly add references for any critical information.

2-      Kindly find if there is (are) any direct reaction with the curcumin and your targeted toxins. You focused only on the role of the immunity? Kindly revise carefully the literatures.

3-      Kindly, rephrase carefully the sentences highlighted with yellow.

In the result part

4-      Kindly remove the last sentence “This section may be divided by subheadings. It should provide a concise and precise description of the experimental results, their interpretation, as well as the experimental conclusions that can be drawn.”

In the discussion part

5-      Kindly avoid reuse of full name with abbreviation described before. Kindly do that in all the manuscript and in the previous sections if any found.

6-      Kindly, add reference(s) to support any information based of previous finding.

7-      I am sure that the words you used did not cover the all the effects that can be happened because of the use of the DON and AFB1. You used to use “combined toxicity of DON and AFB1”.

Kindly find a better way to solve this issue. I suggest, to say “using DON and AFB1 combination” or the like.

In the material and method section

8-      Kindly add more details concerning the animal feeding conditions, water etc., or write using standard criteria following (reference).

Kindly remove the un used section “Patents”

Additional suggestions and remarks

9-      Add references everywhere when there is a need for that.

10-   Concern other studies and compare your data with similar ones even with different compounds.

11-   Avoid re-abbreviate the words, us italic when you use scientific words, describe companies etc.

with my pleasure
